# Depression and physical multimorbidity: A cohort study of physical health condition accrual in UK Biobank

Kelly J. Fleetwood[1☉*], Bruce Guthrie[2☉], Caroline A. Jackson[1], Paul A. T. Kelly[3], Stewart W. Mercer[1], Daniel R. Morales[4], John D. Norrie[1¤], Daniel J. Smith[5], Cathie Sudlow[1,6], Regina Prigge[1]

1 Usher Institute, University of Edinburgh, Edinburgh, United Kingdom, 2 Advanced Care Research Centre, Usher Institute, University of Edinburgh, Edinburgh, United Kingdom, 3 Public Member of Study Advisory Board, Edinburgh, United Kingdom, 4 Division of Population Health and Genomics, University of Dundee, Dundee, United Kingdom, 5 Centre for Clinical Brain Sciences, University of Edinburgh, Edinburgh, United Kingdom, 6 Health Data Research United Kingdom, London, United Kingdom

☉ These authors contributed equally to this work.
¤ Current address: Queen's University, Belfast, United Kingdom
* kelly.fleetwood@ed.ac.uk

## Abstract

### Background

Depression is associated with a range of adverse physical health outcomes. We aimed to quantify the association between depression and the subsequent rate of accrual of long-term physical health conditions in middle and older age.

### Methods and findings

We included 172,556 participants from the UK Biobank (UKB) cohort study, aged 40–71 years old at baseline assessment (2006–2010), who had linked primary care data available. Using self-report, primary care, hospital admission, cancer registry, and death records, we ascertained 69 long-term physical health conditions at both UKB baseline assessment and during a mean follow-up of 6.9 years. We used quasi-Poisson models to estimate associations between history of depression at baseline and subsequent rate of physical condition accrual. Within our cohort, 30,770 (17.8%) had a history of depression. Compared to those without depression, participants with depression had more physical conditions at baseline (mean 2.9 [SD 2.3] versus 2.1 [SD 1.9]) and accrued additional physical conditions at a faster rate (mean 0.20 versus 0.16 additional conditions/year during follow-up). After adjustment for age and sex, participants with depression accrued physical morbidities at a faster rate than those without depression (RR 1.32, 95% confidence interval [CI] [1.31, 1.34]). After adjustment for all sociodemographic characteristics, the rate of condition accrual remained higher in those with versus without depression (RR 1.30, 95% CI [1.28, 1.32]). This association attenuated but remained statistically significant after additional adjustment for baseline condition count and social/lifestyle factors (RR 1.10, 95% CI [1.09, 1.12]). The main limitation of this study is healthy volunteer selection bias, which may limit generalisability of findings to the wider population.

**Data availability statement:** This study was conducted using data from the UK Biobank (https://www.ukbiobank.ac.uk/). Researchers can apply to access the UK Biobank data for health research in the public interest. The code lists used in this study are available from https://github.com/rprigge-uoe/mltc-codelists.

**Funding:** This work was funded by the Medical Research Council (https://www.ukri.org/councils/mrc/)/National Institute for Health Research (https://www.nihr.ac.uk/) (MC/S028013) (BG [principal investigator]; CS, JN, SM, CJ, DM, DS [co-investigators]). The funders of the study had no role in study design, data collection, data analysis, data interpretation, or writing of the report.

**Competing interests:** The authors have declared that no competing interests exist.

**Abbreviations:** BMI, body mass index; CI, confidence interval; CTV3, Clinical Terms V3; HDL, high-density lipoprotein; RR, rate ratio; SBP, systolic blood pressure; MICE, multiple imputation by chained equations; UKB, UK Biobank.

## Conclusions

Middle-aged and older adults with a history of depression have more long-term physical health conditions at baseline and accrue additional physical conditions at a faster rate than those without a history of depression. Our findings highlight the importance of integrated approaches to managing both mental and physical health outcomes.

## Author summary

### Why was this study done?

- Mood disorders like depression are increasingly viewed as whole-body conditions affecting multiple systems across the brain and body.

- People with depression are more likely to have long-term physical health conditions, such as diabetes and arthritis, than people without depression.

- Previous studies have compared people with and without depression in terms of how many long-term physical health conditions they develop over time, however, most studies count 15 or fewer conditions, whereas recent research recommends counting more than 50 conditions.

### What did the researchers do and find?

- We used data from over 170,000 people in middle and older age who participated in the UK Biobank study.

- At the start of the study, 18% of the group had previously been diagnosed with depression.

- We followed up the group for an average of 7 years after the start of the study, using general practitioner and hospital data to identify new diagnoses of 69 long-term physical health conditions.

- At the start of the study, people without a previous diagnosis of depression had an average of 2 long-term physical health conditions, whilst people with a previous diagnosis of depression had an average of 3 such conditions.

- On average, based on people who were the same age and sex, people with a previous diagnosis of depression gained long-term physical health conditions at a 30% faster rate than people without a previous diagnosis of depression.

### What do these findings mean?

- A previous diagnosis of depression is a marker of risk for the subsequent development of long-term physical health conditions during middle and older age.

- Existing healthcare systems are largely designed to treat individual conditions, instead of individual people with multiple conditions, and they especially struggle to treat people with both physical and mental health conditions.

- We need healthcare services to take an integrated approach to caring for people who have both depression (or other mental health conditions) and long-term physical health conditions.

## Introduction

Depression is the most common mental health condition. Over 300 million people worldwide, approximately 1 in 23 of the population, live with depression [1,2]. Prevalence of depression is higher in women than men and most common in adults aged 55–74 years [1]. Depression, irrespective of severity, is associated with increased mortality risk [3]. This is largely due to poorer physical health, particularly a higher risk of type 2 diabetes and cardiovascular disease [4,5].

Multimorbidity is defined as the coexistence of two or more long-term health conditions, with physical multimorbidity specifically referring to the coexistence of two or more long-term physical health conditions. Although estimates of multimorbidity prevalence differ due to large variations in the number of conditions counted [6], multimorbidity increases rapidly with age and is the norm in older people. Health and social care systems are challenged by the increasing prevalence of multimorbidity, driven by population aging, improved survival from acute conditions, and increasing incidence of chronic conditions such as type 2 diabetes [7]. However, the role of depression in multimorbidity is understudied, with only half of multimorbidity studies including depression in their condition count [6]. Whilst depression has been shown to be associated with increased risk of various individual physical health conditions, less is known about its role as a risk factor for physical multimorbidity.

Most previous research on depression and physical multimorbidity has been cross-sectional, with studies indicating that approximately one-fifth of people with any physical condition also have depression, and that the prevalence of depression increases with the number of physical conditions [8]. However, cross-sectional studies are difficult to interpret given a plausible bidirectional relationship between depression and physical health [9].

Previous cohort studies have generally reported an association between baseline depression and accrual of subsequent physical conditions [9–16]. However, these studies have a number of limitations. Most measured physical condition accrual using intermittent surveys where loss to follow-up may be informative because participants who are unwell are more likely to die or drop out [9,10,13,15,17]. Furthermore, many studies measured physical multimorbidity using between three and 15 conditions [9–15,17], whereas a recent Delphi study recommended over 50 conditions for inclusion in multimorbidity measures [18]. Further limitations of existing studies are their relatively short follow-up, with some studies having only three or four years follow-up [9,13,17], and limited adjustment for potential confounders.

We therefore aimed to quantify the association between history of depression and the subsequent accrual of 69 long-term physical health conditions in middle and older age, using electronic health records to identify the incidence of physical conditions during an average of seven years follow-up.

## Methods

We have reported this study in accordance with the STROBE checklist [19] (S1 Checklist). This study is part of a wider project using UK Biobank (UKB) data to investigate relationships between depression and multimorbidity. Our grant proposal and our application to UKB for the wider project included a broad overview of the plans for this study, but we did not publish a prospective analysis plan.

### Study design and participants

The UKB is a cohort study of half a million middle-aged and older adults with information on a wide range of health conditions, sociodemographic, lifestyle, and social factors [20]. People registered with a general practitioner in England, Scotland, or Wales were invited

to participate, with baseline assessment conducted between 2006 and 2010 [21]. Baseline assessment included a touch-screen questionnaire, verbal interview, and physical measurements [20]. Participants provided written informed consent for follow-up through linkage to national datasets, including primary care, hospital, cancer registry, and death records. UKB has ethical approval from the NHS North West Research Ethics Committee (reference: 21/NW/0157).

In order to appropriately evaluate a broad range of long-term physical health conditions, our study population was derived from UKB participants with linked primary care data (approximately 45% of the UKB cohort) [22]. Linked primary care data were available from Scotland, Wales and practices in England that used either the TPP or Vision practice management systems. We included participants with a continuous primary care record (no gaps of >90 days between practice registrations) from at least a year before their baseline assessment to at least one day after baseline assessment [23]. We excluded the small proportion of primary care records from the UKB extract of the Vision practice management system in England because this linked dataset is missing records from people who died before data extraction. We also excluded participants who withdrew from the study.

### Linked electronic health records

In addition to the primary care records, all UKB participants are linked to hospital, cancer registry, and death records from England, Scotland, and Wales. Records are available for different dates in each country, as described in Table A in S1 Text.

We defined conditions at baseline using all primary care records up to and including the date of the participant's baseline assessment. Primary care records transfer between practices in the UK when a patient moves, and so should capture an individual's entire medical history. Cancer registry and hospital records were available from different dates for England, Wales, and Scotland, with a minimum of eight years of records prior to the baseline assessments (Table A in S1 Text). To ensure consistency of look-back period across the secondary care data sources, we defined conditions at baseline for each participant using cancer registry and hospital records from the eight years up to and including their baseline assessment date.

During the follow-up period, both primary care and cancer registry records are available up to at least 2016, with hospital and death records available for longer (Table A in S1 Text). Participants were therefore followed up to the earliest of death, end of continuous primary care record, or end of cancer registry follow-up.

### Outcome

We selected 69 long-term physical health conditions relevant to middle-aged and older adults and recommended for multimorbidity research (Table B in S1 Text) [18,24]. We use the term condition broadly, with some conditions including more than one disease. For example, the condition 'solid organ malignancies' includes both primary and secondary malignancies. This approach reduces double counting of diagnoses that evolve or are related (e.g., for people whose cancer has metastasised).

For each participant, we identified long-term physical health conditions using information collected from the participant at baseline assessment and from primary care, hospital, cancer registry, and death records. We identified conditions from primary care records using Read V2 and Clinical Terms V3 (CTV3) codes, from hospital records using ICD-10 codes and OPCS-4 procedure codes, and from cancer registry and death records using ICD-10 codes. All code lists are available in our GitHub repository (https://github.com/rprigge-uoe/mltc-codelists), with a more detailed description of our approach available in the accompanying manuscript [22].

Conditions were defined as prevalent at each participant's date of UKB baseline assessment if the condition was self-reported at the baseline assessment or the first linked record was on or before the date of the baseline assessment, with the exception of five conditions present from birth (Table B in S1 Text) which were always considered prevalent regardless of the timing of the first record. We defined incident conditions as those where the first record was after the date of baseline assessment. The outcome was the count of incident long-term physical health conditions during follow-up. Since we focused on long-term health conditions, our analyses did not allow for recovery or remission from conditions. Furthermore, such information is not well recorded in the electronic health records.

## Depression

Our exposure is history of depression at baseline: defined as a diagnosis of depression at any time before or on the participant's UKB baseline assessment date. In the United Kingdom, depression is most often diagnosed in primary care, and primary care records should capture a participant's entire medical history, however, in order to capture as many depression cases as possible, we defined history of depression based on multiple sources. We identified diagnoses of depression from linked primary care or hospital records, or self-reported depression in response to the baseline assessment question "Has a doctor ever told you that you have had any other serious medical conditions or disabilities?". Code lists for depression (Read V2 and CTV3 codes for primary care records, and ICD-10 codes for hospital records) are available in our GitHub repository (https://github.com/rprigge-uoe/mltc-codelists).

## Covariates

Age and sex were ascertained from recruitment data and optionally updated by participants at the baseline assessment. We categorised self-reported ethnicity into five groups (Black, mixed, South Asian, White, and any other ethnic group) [25]. Country of residence (England, Wales, or Scotland) and area-based deprivation, measured by the Townsend Deprivation Index (in deciles of the whole UKB cohort) [26], were derived from participants' home addresses at baseline.

For each participant, we calculated a total (prevalent) condition count at their UKB baseline assessment date. This count included the 69 physical health conditions used in the outcome and additionally 10 non-depression mental health conditions (Table C in S1 Text). We ascertained these mental health conditions from baseline assessment, primary care and hospital data using the same approach that we used to ascertain the physical health conditions.

We obtained information on stressful life events, loneliness, multisite pain, sleep, smoking, alcohol intake frequency, and frailty (weight loss, slow walking speed, weak grip strength, and low physical activity) from the baseline assessment touchscreen questionnaire (Table D in S1 Text). We obtained body mass index (BMI), cholesterol:HDL ratio, HbA1c, and systolic blood pressure (SBP) from laboratory tests or measurements taken during the baseline assessment.

## Statistical analysis

Our aim was to describe to what extent the rate of accrual of long-term physical health conditions differs between people with and without depression.

To provide a simple visualisation of the accrual of long-term physical health conditions, we plotted the cumulative mean number of conditions per participant from baseline to end of follow-up by age, sex and history of depression at baseline. The cumulative mean at each time point was based on participants followed up until at least that time point.

We then compared the rate of long-term physical health condition accrual during follow-up between participants with and without a history of depression at baseline using quasi-Poisson models, which allow for over-dispersion in the outcome. We accounted for each participant's follow-up time by including the log follow-up time as an offset term in the models. We estimated the unadjusted association between history of depression at baseline and the count of incident physical conditions, then adjusted for age at baseline and sex, followed by additional sociodemographic characteristics (ethnicity, country of residence at baseline, and area-based deprivation). Finally, we fitted a fully adjusted model which additionally accounted for clinical, social, and lifestyle factors (baseline condition count, count of stressful life events, loneliness, multisite pain, sleep, smoking, alcohol intake frequency, BMI, cholesterol:HDL ratio, HbA1c, SBP, and four markers of frailty: weight loss, slow walking speed, weak grip strength, and low physical activity). Age, count of physical morbidities at baseline, cholesterol:HDL ratio, and SBP were included in the models as continuous variables, each with a linear term and additionally a quadratic term where this improved the fit. Age was scaled by subtracting the mean age and dividing by 10. Count of morbidities at baseline was centred by subtracting the mean. We log transformed cholesterol:HDL ratio and SBP to improve the normality of their distributions (in preparation for the multiple imputation, see below), then centred the logged values by subtracting their mean. All other variables were categorical.

We used multiple imputation by chained equations (MICE) to account for missing data in 17 covariates, assuming that data was missing at random. Most covariates were missing for less than 2% of participants; however, the cholesterol:HDL ratio and HbA1c were missing for 14.2% and 6.1%, respectively (mostly due to assay problems). The imputation models included the outcome and all covariates (including the quadratic terms for the continuous covariates). They did not include any auxiliary variables because the covariates already covered a broad set of sociodemographic, clinical, social and lifestyle factors. Overall, 23.3% of our cohort had at least one missing covariate. Hence, we conducted 25 imputations [27]. Each imputed dataset was analysed and we used Rubin's rules to pool the results [28]. We also conducted a complete case analysis and compared the results to those of the multiple imputation.

All analysis was conducted in R version 4.0.0 or above [29]. We used the mice package version 3.16.0 to perform multiple imputation [30].

## Results

We included 172,556 UKB participants in our cohort (Fig 1). Of these, 30,770 (17.8%) had a history of depression at baseline. History of depression at baseline was predominately identified from primary care records; 90.7% of cases had a primary care diagnosis on or before baseline, 4.2% had a hospital diagnosis on or before baseline, and 31.5% of cases self-reported a previous diagnosis of depression (Fig A in S1 Text). Mean follow-up was 6.9 (SD 1.9) years (Table 1). The mean age at baseline was 57 (SD 8) years. Two-thirds of participants with a history of depression were women, compared to half of participants without a history of depression. The majority of participants were White and from England. Participants with a history of depression more commonly lived in more deprived areas, and on average had 1.3 more long-term health conditions (0.8 more physical health conditions; 0.4 more mental health conditions) at baseline than those without a history of depression. The three most common physical health conditions present at UKB baseline assessment were hypertension (33.4% of people with a history of depression versus 29.6% of people without a history of depression), allergic and chronic rhinitis (32.4% versus 26.7%), and osteoarthritis (27.0% versus 19.1%) (Tables E and F in S1 Text). The most common comorbid mental health conditions present at baseline were anxiety (41.8% versus 6.3%) and alcohol misuse (4.1% versus 1.7%) (Table G in S1 Text). At baseline, chronic multisite pain, sleeplessness, stressful life events, loneliness,

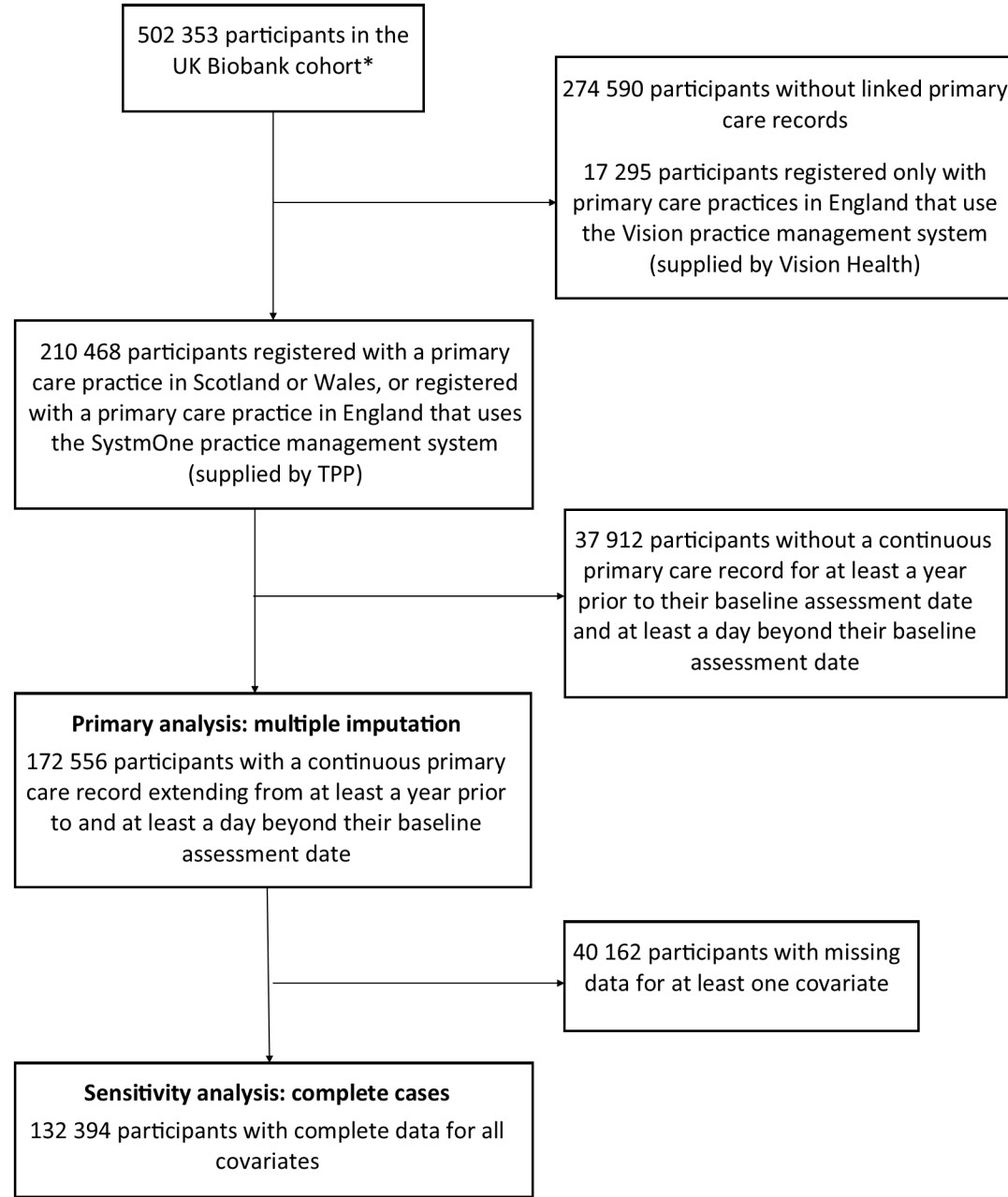

**Fig 1. Flow diagram for the study cohort.** *Excluding participants who withdrew permission for their data to be included in research before 13 October 2023. UK, United Kingdom.

smoking and obesity were all more common amongst participants with a history of depression versus those without. Participants with a history of depression were more likely to abstain from alcohol, or only drink alcohol occasionally. Cholesterol:HDL ratio and HbA1c were marginally higher in participants with a history of depression, and SBP was marginally lower, although absolute differences were small. Participants with a history of depression also had more markers of frailty, including weight loss, slow walking speed, weak grip strength, and low physical activity.

**Table 1. Baseline characteristics of included participants by history of depression at baseline.**

| | History of depression at baseline | |
|---|---|---|
| | Depression (N = 30,770) | No depression (N = 141,786) |
| Follow-up (years) | 6.7 (2.0) | 6.9 (1.9) |
| Age (years) | 56.3 (7.9) | 56.8 (8.0) |
| Sex: Female | 20,592 (66.9%) | 73,431 (51.8%) |
| Ethnicity | | |
| Black | 190 (0.6%) | 1,355 (1·0%) |
| Mixed | 179 (0.6%) | 610 (0.4%) |
| South Asian | 363 (1.2%) | 2,440 (1.7%) |
| White | 29,620 (96.3%) | 135,086 (95.3%) |
| Other ethnic group | 277 (0.9%) | 1,693 (1.2%) |
| Missing | 141 (0.5%) | 602 (0.4%) |
| Country of residence | | |
| England | 23,709 (77.1%) | 107,911 (76.1%) |
| Scotland | 3,617 (11.8%) | 18,497 (13.0%) |
| Wales | 3,429 (11.1%) | 15,335 (10.8%) |
| Missing | 15 (0.0%) | 43 (0.0%) |
| Area-based deprivation (Townsend Deprivation Index quintile*) | | |
| 1 (least deprived) | 5,356 (17.4%) | 29,876 (21.1%) |
| 2 | 5,581 (18.1%) | 29,263 (20.6%) |
| 3 | 6,135 (19.9%) | 29,869 (21.1%) |
| 4 | 6,431 (20.9%) | 28,162 (19.9%) |
| 5 (most deprived) | 7,220 (23.5%) | 24,457 (17.2%) |
| Missing | 47 (0.2%) | 159 (0.1%) |
| Number of prevalent morbidities | 3.5 (2.4) | 2.2 (1.9) |
| Physical | 2.9 (2.3) | 2.1 (1.9) |
| Mental | 0.5 (0.6) | 0.1 (0.3) |
| Chronic multisite pain | | |
| No | 20,729 (67.4%) | 115,487 (81.5%) |
| Yes | 9,928 (32.3%) | 25,755 (18.2%) |
| Missing | 113 (0.4%) | 544 (0.4%) |
| Sleeplessness/insomnia | | |
| Never/rarely | 4,747 (15.4%) | 36,011 (25.4%) |
| Sometimes | 13,820 (44.9%) | 68,121 (48.0%) |
| Usually | 12,122 (39.4%) | 37,337 (26.3%) |
| Missing | 81 (0.3%) | 317 (0.2%) |
| Stressful life events (count, excluding own illness/injury/assault) | | |
| 0 | 15,910 (51.7%) | 88,353 (62.3%) |
| 1 | 10,439 (33.9%) | 41,529 (29.3%) |
| 2 or more | 4,097 (13.3%) | 10,794 (7.6%) |
| Missing | 324 (1.1%) | 1,110 (0.8%) |
| Loneliness | | |
| No | 19,298 (62.7%) | 119,240 (84.1%) |
| Yes | 10,711 (34.8%) | 20,169 (14.2%) |
| Missing | 761 (2.5%) | 2,377 (1.7%) |
| Smoking status | | |
| Never | 15,116 (49.1%) | 79,952 (56.4%) |
| Previous | 10,972 (35.7%) | 48,005 (33.9%) |
| Current | 4,524 (14.7%) | 13,128 (9.3%) |
| Missing | 158 (0.5%) | 701 (0.5%) |

*(Continued)*

**Table 1.** (Continued)

| | History of depression at baseline | |
|---|---|---|
| | Depression (N = 30,770) | No depression (N = 141,786) |
| Alcohol intake | | |
| Daily or almost daily | 5,411 (17.6%) | 28,329 (20.0%) |
| Three or four times a week | 5,832 (19.0%) | 34,203 (24.1%) |
| Once or twice a week | 7,629 (24.8%) | 38,156 (26.9% |
| One to three times a month | 3,850 (12.5%) | 15,543 (11.0%) |
| Special occasions only | 4,566 (14.8%) | 14,777 (10.4%) |
| Never | 3,394 (11.0%) | 10,493 (7.4%) |
| Missing | 88 (0.3%) | 285 (0.2%) |
| BMI (kg/m²) | | |
| <25 | 8,873 (28.8%) | 46,121 (32.5%) |
| 25–29.9 | 12,434 (40.4%) | 60,998 (43.0%) |
| 30–34.9 | 6,143 (20.0%) | 24,737 (17.4%) |
| ≥35 | 3,121 (10.1%) | 9,132 (6.4%) |
| Missing | 199 (0.6%) | 798 (0.6%) |
| Cholesterol:HDL ratio | | |
| Mean (SD) | 4.2 (1.1) | 4.1 (1.1) |
| Missing | 4,530 (14.7%) | 19,913 (14.0%) |
| HbA1c (mmol/mol) | | |
| <32 | 4,949 (16.1%) | 24,394 (17.2%) |
| 32–34 | 4,909 (16.0%) | 23,667 (16.7%) |
| 34–36 | 5,934 (19.3%) | 28,770 (20.3%) |
| 36–38 | 5,237 (17.0%) | 23,703 (16.7%) |
| ≥38 | 7,820 (25.4%) | 32,603 (23.0%) |
| Missing | 1,921 (6.2%) | 8,649 (6.1%) |
| Systolic blood pressure (mmHg) | | |
| Mean (SD) | 136.2 (18.4) | 139.2 (18.7) |
| Missing | 160 (0.5%) | 451 (0.3%) |
| Weight loss | | |
| No | 24,752 (80.4%) | 118,619 (83.7%) |
| Yes | 5,433 (17.7%) | 20,545 (14.5%) |
| Missing | 585 (1.9%) | 2,622 (1.8%) |
| Slow walking speed | | |
| No | 26,191 (85.1%) | 130,868 (92.3%) |
| Yes | 4,163 (13.5%) | 9,878 (7.0%) |
| Missing | 416 (1.4%) | 1,040 (0.7%) |
| Weak grip strength | | |
| No | 21,875 (71.1%) | 109,169 (77.0%) |
| Yes | 8,641 (28.1%) | 31,647 (22.3%) |
| Missing | 254 (0.8%) | 970 (0.7%) |
| Low physical activity | | |
| No | 25,934 (84.3%) | 126,933 (89.5%) |
| Yes | 4,441 (14.4%) | 13,654 (9.6%) |
| Missing | 395 (1.3%) | 1,199 (0.8%) |

Data are mean (SD) or n (%).

*The statistical models adjust for Townsend Deprivation Index decile, however, quintiles are shown here for conciseness.

BMI, Body Mass Index; HbA1c, glycated haemoglobin; HDL, high density lipoprotein.

During follow-up, participants with a history of depression accrued an average of 0.20 additional physical health conditions per year compared to 0.16 conditions per year for participants without a history of depression (Fig 2). The three most common incident physical health conditions during follow-up in both participants with and without a history of depression were osteoarthritis (15.7% of people with depression, but without osteoarthritis at baseline versus 12.5% of people without depression or osteoarthritis at baseline), hypertension (12.9% versus 12.0%) and gastro-oesophageal reflux disease (and similar) (13.8% versus 9.6%), (Table E in S1 Text).

Based on all participants in our cohort, in the unadjusted model, people with a history of depression accrued new long-term physical health conditions at a faster rate than people without a history of depression (rate ratio [RR] 1.25, 95% confidence interval [CI] [1.23, 1.27]). After adjustment for age and sex, the rate of condition accrual remained higher in people with versus without a history of depression (RR 1.32, 95% CI [1.31, 1.34]) (Table 2). Additional adjustment for sociodemographic factors only slightly attenuated the effect estimate (RR 1.30, 95% CI [1.28, 1.32]). In the fully adjusted model, which included potential mediators, a history of depression was associated with a marginally increased rate of morbidity accrual (RR 1.10, 95% CI [1.09, 1.12]).

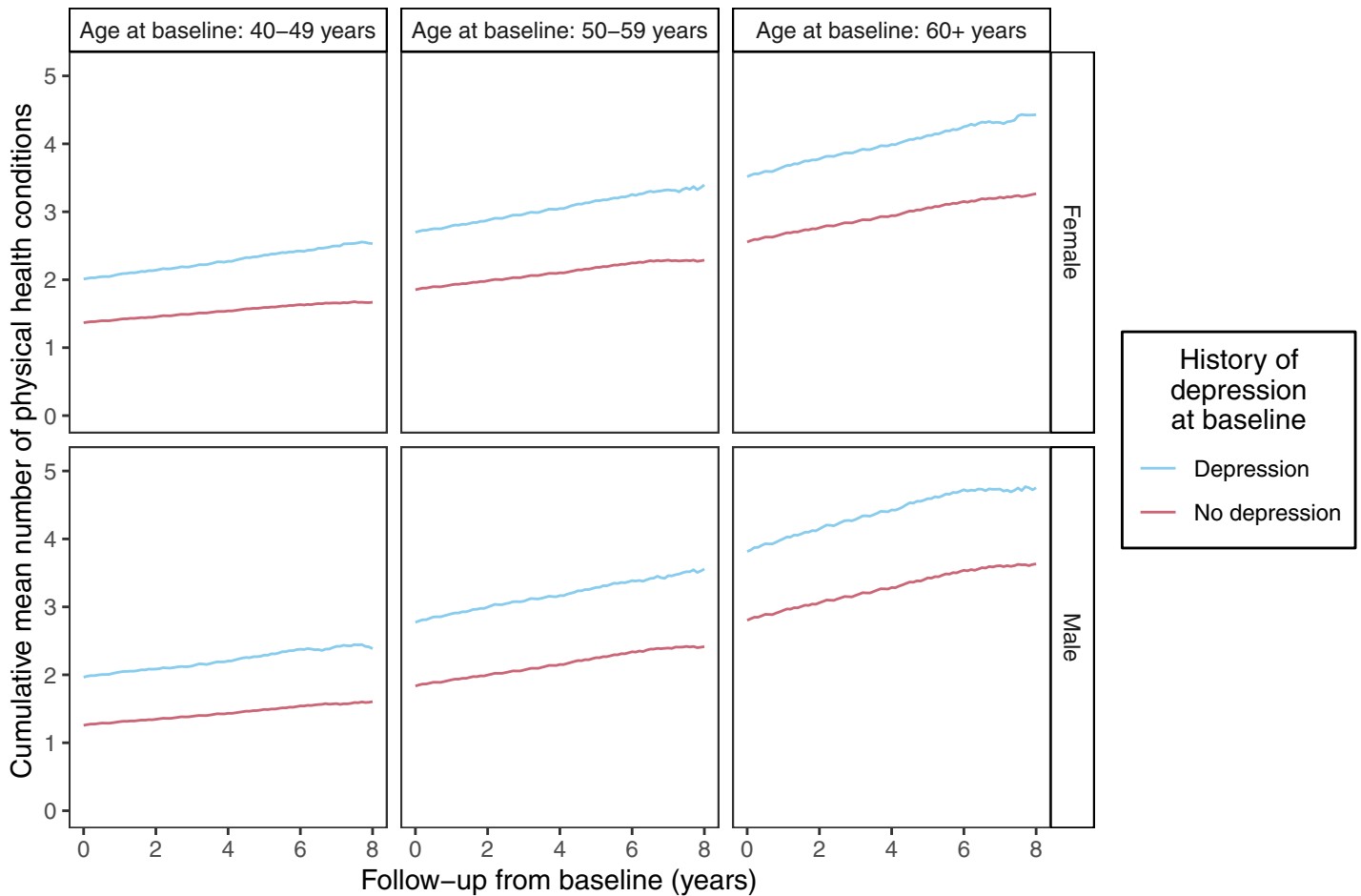

**Fig 2. Cumulative mean number of long-term physical health conditions at baseline and during follow-up\*, stratified by history of depression at baseline, age at baseline and sex (n = 172,556).** \*The cumulative mean at each time point is based on participants followed up until at least that time point.

**Table 2. Rate ratios for the association of history of depression at baseline, sociodemographic, social, lifestyle and clinical factors with physical health condition accrual during follow-up.**

| | | Rate ratio (95% CI)* | | |
| --- | --- | --- | --- | --- |
| | | Adjusted for age and sex | Adjusted for sociodemographic characteristics | Fully adjusted |
| History of depression at baseline | | 1.32 (1.31, 1.34) | 1.30 (1.28, 1.32) | 1.10 (1.09, 1.12) |
| Age† | Age (RR for each additional 10 years of age) | 1.58 (1.57, 1.59) | 1.59 (1.58, 1.61) | 1.43 (1.41, 1.44) |
| Sex (ref: Male) | Female | 0.80 (0.79, 0.81) | 0.81 (0.80, 0.82) | 0.82 (0.81, 0.83) |
| Socioeconomic status Townsend deprivation decile (ref: 1, least deprived) | 2 | | 0.99 (0.96, 1.01) | 0.98 (0.96, 1.01) |
| | 3 | | 1.01 (0.98, 1.03) | 0.99 (0.97, 1.01) |
| | 4 | | 1.04 (1.01, 1.07) | 1.02 (0.99, 1.04) |
| | 5 | | 1.06 (1.03, 1.08) | 1.02 (0.99, 1.04) |
| | 6 | | 1.06 (1.03, 1.08) | 1.00 (0.98, 1.03) |
| | 7 | | 1.10 (1.07, 1.12) | 1.01 (0.99, 1.04) |
| | 8 | | 1.13 (1.11, 1.16) | 1.02 (1.00, 1.05) |
| | 9 | | 1.21 (1.18, 1.24) | 1.04 (1.02, 1.07) |
| | 10 (most deprived) | | 1.35 (1.32, 1.39) | 1.07 (1.04, 1.10) |
| Ethnicity (ref: White) | Black | | 1.20 (1.13, 1.28) | 1.08 (1.02, 1.14) |
| | Mixed | | 1.11 (1.02, 1.21) | 1.06 (0.97, 1.15) |
| | South Asian | | 1.33 (1.28, 1.39) | 1.16 (1.11, 1.20) |
| | Other ethnic group | | 1.10 (1.04, 1.16) | 1.03 (0.98, 1.09) |
| Country of residence at baseline (ref: England) | Scotland | | 0.81 (0.80, 0.82) | 0.82 (0.81, 0.84) |
| | Wales | | 0.99 (0.98, 1.01) | 0.95 (0.93, 0.97) |
| Morbidities at baseline† | Count | | | 1.06 (1.06, 1.07) |
| | Count-squared | | | 0.997 (0.996, 0.997) |
| Smoking (ref: never) | Previous | | | 1.09 (1.08, 1.11) |
| | Current | | | 1.25 (1.23, 1.28) |
| Alcohol intake (ref: daily or almost daily) | 3–4 times a week | | | 0.97 (0.95, 0.99) |
| | 1–2 times a week | | | 1.00 (0.98, 1.02) |
| | 1–3 times a month | | | 1.01 (0.99, 1.03) |
| | Special occasions | | | 1.05 (1.03, 1.07) |
| | Never | | | 1.08 (1.06, 1.10) |
| BMI (kg/m²) (ref: <25) | 25–29.9 | | | 1.10 (1.08, 1.11) |
| | 30–34.9 | | | 1.20 (1.18, 1.22) |
| | ≥35 | | | 1.33 (1.30, 1.36) |
| SBP (mmHg)† | log(SBP) | | | 1.42 (1.35, 1.48) |
| | (log(SBP))-squared | | | 3.20 (2.59, 3.95) |
| Cholesterol:HDL ratio† | log(ratio) | | | 1.06 (1.03, 1.08) |
| | (log(ratio))-squared | | | 1.24 (1.17, 1.32) |
| HbA1c (mmol/mol) (ref: <32) | 32–34 | | | 0.99 (0.97, 1.01) |
| | 34–36 | | | 1.02 (1.00, 1.04) |
| | 36–38 | | | 1.04 (1.02, 1.06) |
| | ≥38 | | | 1.21 (1.19, 1.23) |
| Sleeplessness/insomnia (ref: never/rarely) | Sometimes | | | 1.04 (1.02, 1.05) |
| | Usually | | | 1.09 (1.07, 1.11) |
| Count of stressful life events (ref: 0) | 1 | | | 1.03 (1.02, 1.04) |
| | 2 or more | | | 1.06 (1.04, 1.08) |
| Chronic multisite pain (ref: no) | Yes | | | 1.15 (1.14, 1.17) |
| Loneliness (ref: no) | Yes | | | 1.06 (1.05, 1.08) |

*(Continued)*

**Table 2.** (Continued)

| | | Rate ratio (95% CI)* | | |
|---|---|---|---|---|
| | | Adjusted for age and sex | Adjusted for sociodemographic characteristics | Fully adjusted |
| Weight loss (ref: no) | Yes | | | 1.04 (1.03, 1.06) |
| Slow walking speed (ref: no) | Yes | | | 1.12 (1.10, 1.14) |
| Weak grip strength (ref: no) | Yes | | | 1.05 (1.03, 1.06) |
| Low physical activity (ref: no) | Yes | | | 1.08 (1.06, 1.10) |

*Results of Poisson models, multiple imputation was used to account for missing covariates in the model adjusted for sociodemographic characteristics and the fully adjusted model.

† For each quantitative covariate, we scaled the values by subtracting their mean.

BMI, body mass index; HbA1c, glycated haemoglobin; HDL, high-density lipoprotein; SBP, systolic blood pressure.

For all models, results based on complete case analysis were similar to those based on multiple imputation (Table H in S1 Text).

## Discussion

Compared to people without a history of depression, people with a history of depression had more long-term physical health conditions at baseline, and accrued physical conditions at a faster rate even after accounting for sociodemographic differences. The association attenuated after further adjustment for baseline clinical, social, and lifestyle factors.

A number of other studies have examined the association between depression or depressive symptoms and the accrual of long-term physical health conditions [9–16]. These studies varied in the number of physical health conditions studied, how conditions were followed-up and the length of follow-up, but most also found that depression or depressive symptoms were associated with a higher rate of condition accrual. Using UKB data, Qiao and colleagues (2022) found that baseline depression was associated with higher rates of cardiometabolic multimorbidity (narrowly defined as two or more of type 2 diabetes, stroke, and coronary heart disease) [14]. A large study from Canada of adults without physical morbidity at baseline observed associations between a major depressive episode in the previous 12 months and the development of one or more of 15 conditions in the following 10 years [12]. Whilst a Swedish study of older adults defined physical multimorbidity based on 54 conditions and had a follow-up of 15 years, this study was much smaller than our own with approximately 3,000 participants in total and fewer than 300 with depression [16]. Nevertheless, it also found an association between depression and rate of morbidity accumulation. In contrast, a study from the United States found that adults with baseline bipolar disorder, but not depression, had a higher rate of morbidity accrual compared to adults without a mood disorder [17]. However, the study population was younger than in the present study, with almost half the participants younger than 45 years of age at baseline, and follow-up was only three years. Using data from residents of one US county, Bobo and colleagues (2011) found that women with depression experienced higher rates of accrual of 15 physical conditions across the life span, although associations in men varied by age and by the presence of comorbid anxiety, with a higher rate of condition accrual in some groups of men, but no evidence of association in others [11].

Strengths of this study are the use of a very large cohort with multiple linked electronic health records including primary care data, which is essential for detecting many conditions that are treated mainly or exclusively in primary care, including depression [22]. Our definition of physical multimorbidity included a wide range of long-term physical health conditions based on recent recommendations for conditions to include in multimorbidity research [18].

However, we acknowledge that there is ongoing debate about what conditions should be included in multimorbidity measures, and whether some conditions, such as hypertension, should instead be treated as risk factors. Average follow-up in this study was seven years and used linked electronic health records, including death records, meaning that outcome ascertainment is likely to be better than previous studies which relied on repeat survey data, and are therefore subject to recall and reporting bias, and ascertainment bias due to loss to follow-up [31].

The study also has a number of limitations. Only 5.5% of people invited to join the UK Biobank participated in the baseline assessments, with participants less likely on average to live in deprived areas, and more likely to have better health, compared with the general population [32]. Baseline condition counts and rates of accrual of new conditions are therefore like to underestimate values in the general population. Furthermore, our estimates of the association between history of depression at baseline and physical health condition accrual may not be generalisable to the wider population [33,34]; however, it is reassuring that other multimorbidity associations estimated from the UK Biobank cohort were generally similar to associations estimated from a nationally representative sample [35]. Whilst our partially adjusted models included information on covariates that likely preceded both our exposure and outcome and thus acted as confounding factors, some covariates included in our final model may lie on the causal pathway or have a conceptual overlap with depression. Thus, our final model may have underestimated the strength of the association between depression and physical condition accrual. Our cohort had relatively high rates of missingness for cholesterol:HDL ratio (14.2%) and HbA1c (6.1%), although data for other covariates was near complete. We accounted for missing data using multiple imputation under the assumption that data was missing at random which is plausible since missing laboratory data was primarily due to assay problems after sample collection. Whilst we have examined a broad range of long-term physical health conditions, examining the relationship between depression and specific physical conditions was outside the scope of this study. Finally, we were unable to examine the role of depression remission and relapse and depression severity because such information was not reliably captured by the data.

Middle-aged and older adults with a history of depression have higher prevalence of physical health conditions at baseline, and have an increased rate of physical condition accrual subsequently. The higher rate of accrual is partly driven by differences in modifiable risk factors like smoking, high BMI and low physical activity, meaning that there are potential opportunities for preventive care to improve future health. Multimorbidity challenges existing healthcare because individual needs do not always sit neatly with organisational boundaries, and that is often most true when people have both physical and mental health problems. Better identification and management of depression in physical healthcare is needed, but mental health services also need to involve themselves in supporting their patients to maintain or improve their physical health, in relation to smoking, diet, obesity, and exercise, for example. Relatively intensive collaborative care approaches have been shown to be effective in improving mental and physical outcomes in people with depression and common conditions like diabetes and heart disease but there is a need to develop and evaluate more consistent preventive approaches across all services [36]. Such approaches are likely to be most needed in less affluent areas, with the least affluent developing any multimorbidity 10–15 years earlier than the most affluent, and even larger differences in physical-mental multimorbidity [8,37]. In this study, people living in more deprived areas were more likely to have depression at baseline. This highlights the additional need for better integrated physical-mental healthcare in less affluent areas, which is not reflected in current resource allocation [38].

There are a number of implications for future research. Although people with a history of depression accrued physical conditions at a higher rate than people without, they also already had more physical health conditions at baseline. Life course approaches would be useful to better understand the interplay of depression and physical health at younger ages, and to unpick whether the observed associations are driven by a subset of physical conditions or whether associations are similar for all conditions. Investigating the ordering of subsequent long-term physical health conditions [39] would also be worthwhile, although it is often difficult to establish the order of conditions from electronic health records, especially where multiple diagnoses are recorded on the same day. Similarly, it would be useful to explore whether depression is also associated with subsequent physical condition severity and disease-specific outcomes, as well as generic outcomes like quality of life. We estimated the association between depression and physical condition accrual in a broad cohort, including both men and women, ranging in age from 40 to 71 years at baseline. However, it would also be valuable to explore this association in analyses stratified by key sociodemographic characteristics such as age and sex. Further research should also more carefully explore the causal relationship between depression and the subsequent accrual of long-term physical health conditions in order to identify potential interventions, including whether effective management of depression and/or more assertive management of risk factors for physical disease in people with depression leads to lower rates of physical disease accrual. Further research which stratifies depression by severity and chronicity would be useful to examine if there are particular groups of patients where such interventions might be most effectively targeted. Research is also needed to better understand potential mechanisms, for example, the role of inflammation in the development of both depression and physical health conditions like coronary heart disease.

In the most comprehensive study to date on this topic, to our knowledge, we identified a higher rate of accrual for comorbid physical health problems in people with a history of depression compared to those without. Our findings highlight that depression should be viewed as a 'whole body' condition, as well as the importance of integrated approaches to managing both mental and physical health outcomes.

## Supporting information

**S1 Checklist. STROBE checklist.**
(DOCX)

**S1 Text. Supplementary tables and figures.**
(DOCX)

## Acknowledgments

The study was conducted using the UK Biobank Resource under application number 57213. The authors would like to thank the UK Biobank participants and the UK Biobank staff for their contributions to this study. The authors would also like to thank Pat Watson, a public member of our advisory board, for providing thoughtful feedback throughout our project, Dr Emma Davidson for providing us with unpublished depression code lists and Dr Kristiina Rannikmäe for providing us with stroke, B12 deficiency anaemia, folate deficiency anaemia and iron deficiency anaemia code lists which were unpublished at the time.

## Author contributions

**Conceptualization:** Bruce Guthrie, Caroline A. Jackson, Stewart W. Mercer, Daniel R. Morales, John D. Norrie, Daniel J. Smith, Cathie Sudlow.

**Data curation:** Kelly J. Fleetwood, Regina Prigge.

**Formal analysis:** Kelly J. Fleetwood, Regina Prigge.

**Funding acquisition:** Bruce Guthrie, Caroline A. Jackson, Stewart W. Mercer, Daniel R. Morales, John D. Norrie, Daniel J. Smith, Cathie Sudlow.

**Supervision:** Bruce Guthrie.

**Writing – original draft:** Kelly J. Fleetwood, Bruce Guthrie.

**Writing – review & editing:** Kelly J. Fleetwood, Bruce Guthrie, Caroline A. Jackson, Paul AT Kelly, Stewart W. Mercer, Daniel R. Morales, John D. Norrie, Daniel J. Smith, Cathie Sudlow, Regina Prigge.

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
