## [Editor Report · Decision Letter 0]

9 Aug 2024

Dear Dr Fleetwood, 

Thank you for submitting your manuscript entitled "Depression and multimorbidity: a cohort study of physical health condition accrual in UK Biobank" for consideration by PLOS Medicine.

Your manuscript has now been evaluated by the PLOS Medicine editorial staff and I am writing to let you know that we would like to send your submission out for external peer review.

Please re-submit your manuscript within two working days, i.e. by Aug 13 2024 11:59PM.

Kind regards,

Pippa

Philippa C. Dodd, MBBS MRCP PhD

Senior Editor

PLOS Medicine

pdodd@plos.org

---

## [Decision Letter · Decision Letter 1]

22 Oct 2024

Dear Dr Fleetwood,

Many thanks for submitting your manuscript "Depression and multimorbidity: a cohort study of physical health condition accrual in UK Biobank" (PMEDICINE-D-24-02561R1) to PLOS Medicine. The paper has been reviewed by subject experts and a statistician; their comments are included below and can also be accessed here: [LINK]

As you will see, the reviewers were in agreement that the paper dealt with an important area of research. However, they also raised a number of points that will need addressing. After discussing the paper with the editorial team and an academic editor with relevant expertise, I'm pleased to invite you to revise the paper in response to the reviewers' comments. We plan to send the revised paper to some or all of the original reviewers, and we cannot provide any guarantees at this stage regarding publication.

When you upload your revision, please include a point-by-point response that addresses all of the reviewer and the editorial points, indicating the changes made in the manuscript and either an excerpt of the revised text or the location (eg: page and line number) where each change can be found. Please also be sure to check the general editorial comments at the end of this letter and include these in your point-by-point response. When you resubmit your paper, please include a clean version of the paper as the main article file and a version with changes tracked as a marked-up manuscript. It may also be helpful to check the guidelines for revised papers at http://journals.plos.org/plosmedicine/s/revising-your-manuscript for any that apply to your paper.

We ask that you submit your revision by Nov 12 2024 11:59PM. However, if this deadline is not feasible, please contact me by email, and we can discuss a suitable alternative.

Don't hesitate to contact me directly with any questions (ssunny@plos.org). 

Best regards, 

Syba

Syba Sunny MBBS, MRes, FRCPath

Associate Editor 

PLOS Medicine

ssunny@plos.org

Comments from the academic editor:

The academic editor was very much in favour of your paper and stated the following: ‘this study is one of the most robust in all respects (sample size, measures, longitudinal design etc) to disentangle the complicated relationship between depression and chronic conditions and has important findings for policy and practice’. Nevertheless, he agreed with the reviewer comments and, thus, asks that the authors revise their manuscript accordingly.

Comments from the reviewers: 

Reviewer #1: See attachment

Michael Dewey

Reviewer #2: Review - PMEDICINE-D-24-02561R1

Authors present a study examining the association between history of depression and the accumulation of physical diseases using data from UKB. Using a large study population with a wide age range at baseline (40-71 years), they report a faster rate of disease accumulation in people with a history of depression. These associations are preserved after accounting for ethnicity, area-level deprivation, and country of residence. Subsequent adjustment for clinical, behavioural, and functional indicators, as well as prevalent diseases, reduces the effect of depression history, although it remains. The paper is well-written and the analysis is sound. I have some concerns about conceptualizing depression, lack of age-group stratification given the wide age-span of the sample, and overall innovativeness of the study, given the many studies examining depression-multimorbidity connection out there. 

1. Authors are not consistent in their terminology/conceptualization of depression. In some sections, they speak about prevalent depression (eg., people with/without depression), whereas in most other places, they write "history of depression". In my reading, their operationalization of depression is more consistent with the notion of "depression history", at least when it comes to the self-report item. For the electronic records this is less clear, and more detail is required from authors here. Does depression assessment from the records only refer to the last eight years, or is it possible to gauge the history of depression from these data? Depending on the answer here, the conceptualization of depression diagnosis will vary. Consequently, if one source of depression diagnosis used here refers to history (self-repot), while the other to the last eight years, then combining them into a single measure may not be suitable. Please provide more detail about conceptualizing depression from the different sources and consider sensitivity analysis with alternative operationalizations of depression (using just one source of diagnosis?) to limit misclassification and ensure consistency between operational definition and its conceptualization.

2. What is the overlap between the self-report of depression and the diagnosis from the electronic records?

3. Did the authors consider episodic nature of depression in their operationalization? Is it handled in any way, particularly for data from electronic records? At the very least, this should be discussed in the limitations.

4. Authors are working with a very wide age range (40-71 years at baseline). We know that both multimorbidity and depression are age-sensitive syndromes, increasing in prevalence with age (multimorbidity), or exhibiting a more chronic course and poor prognosis, with an altered clinical presentation (depression). Indeed, some have argued that old-age depression may be conceptually distinct from middle-aged depression, with an outsized role played by somatic and cognitive symptoms (as opposed to affective/low mood ones). Given such age-dependency in both exposure and outcome, I think it is imperative to not lump all ages together, but rather conduct stratified analysis and discuss age-specific findings at length.

5. To a large extent, this also applies to sex. In fact, authors themselves mention sex differences in the introduction, therefore it would only be natural to revisit them later on in the paper.

6. For the operationalization of chronic conditions, I have several points. 1) Did the authors consider dropping hypertension from disease count and re-calculating accrual rates without it? Some in MM literature have argued that hypertension represents a risk factor rather than an overt disease, and given its high prevalence, the inclusion of hypertension could be misleading. 2) I was unclear about the mental conditions other than depression... Were they considered as part of the total count in the accrual analysis? If they were, authors should do a sensitivity analysis without them, considering the comorbidity of mental disorders, particularly depression and anxiety (especially in older adults). 3) please consider providing baseline prevalences of diseases in the Supplementary Table 1.

7. The inclusion of stressful life events, loneliness, pain, and sleep could be problematic, considering their possible conceptual overlap with depression (indeed sleeping difficultly is one of depression symptoms). This is further complicated by the fact they are assessed concurrently with depression. I suggest their inclusion is reconsidered.

8. Indeed, the analysis in which factors on the causal pathway between depression and disease count are considered as a way of explaining away depression's effect is not well justified in the introduction. As I read, the aim seems to be about predictive effects of depression on multimorbidity change, whereby depression and MM are kept temporally and conceptually distinct to strengthen inferences. This analysis, blurs this conceptual separation, and requires better justification.

9. How much of the drop in the effect of depression is attributable to controlling for prevalent conditions?

10. I have doubts about the usefulness of the results in Table 2 according to prevalent/incident individual diseases. To me, this goes back to lacking age-group perspective. Ignoring huge age differences runs the risk of misrepresenting these cross-tabs. Also, in the limitations, authors write that examining the relationship between depression and specific physical conditions was outside the scope of this study. Still, they devoted a lengthy table 2 to individual diseases. I think the paper would benefit more from an explicit age/sex perspective, considering the size of authors' dataset.

11. In the face of lacking age/sex/disease type/depression severity/course perspective, I am left wondering about the place of this study in the field where depression-multimorbidity associations have been extensively researched before (and synthesized in previous reviews, including scoping and systematic ones). Undoubtedly, authors address some previous gaps (primarily study size and depth of multimorbidity assessment), although they are confronted with others such as depression conceptualization issues and non-representativeness of the study population. And given the lack of an "angle" that helps triangulate these findings a specific and well-defined at-risk subpopulation, I am left wondering exactly what new this study brings to the table (or, to rephrase it, whether should it be published in a high-impact journal like PLOS Medicine).

Reviewer #3: This paper tackles a highly relevant topic, investigating the relationship between depression and the accumulation of physical health conditions (multimorbidity) using the UK Biobank cohort. The study is well-structured, leverages a robust dataset, and contributes valuable insights to the growing body of literature on mental health and its intersection with chronic disease. The paper demonstrates methodological rigor and offers important implications for public health practice and clinical care. However, there are several areas where the manuscript can be improved to enhance clarity, rigor, and interpretability of results.

One question relates to the choice of covariates:

The authors include a lot of covariates in their analysis, and the justification for selecting manyof these is not clear. Were they chosen because of theoretical importance? 

Reviewer #4: This study addresses an important research topic on the impact of depression on the subsequent accrual of physical health conditions. The strengths of the work include its large cohort size (172,556) participants and the comprehensive inclusion of conditions in the definition of multimorbidity - which set this paper aside from many of the existing studies in the literature at present. Whilst at face value the analyses is well conducted and presented, in particular, a high quality imputation approach as been implemented - I do have several major concerns regarding the overall analyses and modelling approach which are critical to the overall reliability and quality of the results. 

1. Introduction - paragraph on multimorbidity and depression studies being cross-sectional references many of the key papers, however it is incomplete and could be more balanced therefore. e.g. There are studies which look at disease accrual over time/in sequence (e.g. Hayward et al. Lancet eBioMedicine, 2023 Volume 96, 104792 - disease trajectories) which shows data on the time sequenced accrual of mental health related hospital admissions following cardiovascular diseases and its association with increased risk of death. A more comprehensive review of the literature to highlight the novel contribution of this work therefore is recommended. 

2. Whilst UK Biobank is a fantastic resource - the disadvantages of its use for multimorbidity research must be highlighted up front. The authors use strong language in the introduction to suggest they "robustly" identify physical conditions, and whilst I appreciate they have gone over and above existing studies in their attempts to be much more inclusive and comprehensive in accrual of conditions than others - the level of selection bias and healthy user bias inherent in UK Biobank does limit the robustness of this assessment. Further linked primary care data is available for fewer than 50% of the cohort. Further data on the level of linkage to other data sources do not feature strongly (or at all?) - and these all affect the level of confidence in ascertainment and the equality of ascertainment of both depression and the physical conditions for individuals in the study. Moreover over half the cohort are excluded given the lack of primary care linkage - I realise this is a necessary choice given the data but this does introduce further queries over the representativeness of these data and its conclusions which need to be fully acknowledged as limitations. The language in the intro and discussion need to be appropriately moderated to reflect this also.

Statistical analyses

3. High quality imputation approach has been used (which is rare - and therefore commended)! - the approach follows good practice guidelines for MICE such as including the outcome variable and all covariates and their formulations, as well as a comparison with complete cases. Please state whether any additional auxiliary variables were included in the imputation models or not, and include a detailed description of missing data for all variables (e.g. by an extra column in Table 1). Missing data are reported for some but not all variables in the table - e.g. it is hard to believe there were no missing data at all for BMI and alcohol intake - yet missing data for these are not included in Table 1. 

4. Table 2 presents details of those with and without depression according to the conditions at baseline and those accrued during the follow up period. It is not clear however what the timing of those baseline conditions in relation to depression is? They can be before or after the depression as far as I can tell - which makes interpretation of this table difficult. Could the authors elaborate on this? How does this impact on your results? 

5. "Baseline depression" has been identified as any occurrence of depression recorded during an 8 year medical history - however this is likely to lead to a heterogenous population with differential impact on outcomes for e.g. individuals who had one short episode of depression related to an acute event vs. those with longer term/persistent depression or mental health conditions over the length of follow up. Has this been accounted for as these two scenarios are likely to influence the impact of depression physical health conditions very differently. 

6. Observational studies of this type should at minimum use some form of matching between cohorts to minimise additional sources of bias. I would expect to see either a propensity score analyses, or other form of matching (e.g. risk-set matching approach) minimise bias between the groups (esp. given concerns about ascertainment highlighted below). 

7. Given that ascertainment of depression and physical health conditions will vary hugely between primary, secondary care and the UK biobank baseline data it is important to understand how many cases included in the study were linked with the various datasets and over what time periods these linkages were present. Those with fully linked data may have an increase in physical health conditions recorded and a higher chance of a record for depression vs those who have fewer linked datasets and it would be important to know the degree with which this varies for individuals in the cohort. How does this impact on your results, and has this been accounted for in any way?

8. The use of centring and quadratics in the modelling is a little outdated and requires either further justification plus clear model diagnostics to be presented (e.g. why was age divided by 10 specifically? How much improvement in model fit were you looking for in the quadratics? centring is usually done to aid interpretation, but as these are just included as confounders and should not to be directly assessed (See point later about "table 2 fallacy" - there seems little point). This approach overall is a little inefficient and requires a number of steps and judgements to be made in the process. Accounting for non-linearity in variables is better done via the use of restricted cubic splines which estimates all transformations simultaneously (instead of adding quadratic terms for all predictors and then determining which ones should be kept). This approach would also lead to appropriate confidence intervals of your estimates - which are not influenced by scale of the data and therefore do not require centring.

9. The modelling approach does not appear to take different follow up periods for individuals and any censoring into account (as far as I could discern from the paper). Lack of accounting for this will cause major biases in your model which should not be published unless addressed (e.g. at minimum through inclusion of an appropriate offset term in the Poisson model, but ideally through use of a time to event modelling framework). In the current analyses it is far more likely to identify someone in the "depression" group if they have more follow up time (i.e. there is more opportunity to identify their existing diagnoses) and therefore any comparisons between a depressed and non-depressed group is likely to be inherently biased. In addition to that - the outcome (number of conditions accrued) is heavily influenced by length of follow up time and censoring - which therefore must be addressed in the modelling directly. 

10. The inclusion of a good number of confounders is commended - however - further thought needs to be provided as to why those variables are indeed included. Were other data available and discounted, are all these variables definitely confounders and not mediators? The use of a DAG in this instance of many variables collected for a dataset such as these is highly recommended, which would also identify any further potentially important confounders which were not available to be listed in the discussion. 

11. The authors present their modelling data in such a way that falls foul of the so called "Table 2 Fallacy" - which is when estimates of confounders are presented and interpreted directly as exposures in their own right whilst adjusting for all other confounders. The only relevant estimates to be interpreted and presented for this particular model is the RRs for depression vs no depression. It's fine to include a model with several "layers" of confounders, although STROBE/RECORD guidance recommends the reporting of unadjusted RRs in addition to these also. The RRs for all the confounders do not have / should not be interpreted independently in any way and therefore should not be presented in the table or described as they are in the results section. The only act as confounders in this case. If the authors are additionally interested in the effects of age on morbidity accrual - then a new model should be generated with age as the main exposure and a bespoke set of confounders relevant to the age-outcome relationship should be identified, and this should also be added to the original aims of the presented work. 

12. Figure 2 shows a plot of mean number of long term conditions - this is a simplistic view of these data which may be biased by not accounting for differential follow up time for individuals contributing to these data, and confidence intervals should be displayed for these curves also. The use of formal time to event analyses accounting directly for censoring and length of follow up time is advised here (e.g. calculating cumulative incidence functions) and displayed with confidence intervals. It is also good practice to list the numbers at risk in time intervals (e.g. for each year of follow up) for each of these plots so that the accuracy of the longer term data (which is a stated strength of this paper). There will be fewer individuals with longer term follow up data - and the drop off rate of this should be clearly presented. 

13. Aside from the table 2 fallacy as already discussed - the strength of association cannot be compared in this way with respect to size of estimate; therefore this statement and others similar throughout the manuscript should be removed : " The association between depression and the accrual of physical conditions was of a similar magnitude to associations between other characteristics and condition accrual, although weaker than associations with age, smoking, and BMI in the fully adjusted model".

---

* Please upload any figures associated with your paper as individual TIF or EPS files with 300dpi resolution at resubmission; please read our figure guidelines for more information on our requirements: http://journals.plos.org/plosmedicine/s/figures. While revising your submission, please upload your figure files to the PACE digital diagnostic tool, https://pacev2.apexcovantage.com/. PACE helps ensure that figures meet PLOS requirements. To use PACE, you must first register as a user. Then, login and navigate to the UPLOAD tab, where you will find detailed instructions on how to use the tool. If you encounter any issues or have any questions when using PACE, please email us at PLOSMedicine@plos.org.

* Thank you for including a completed STROBE checklist. Please move this document to ‘Supporting Information’ and add the following statement, or similar, to the Methods: "This study is reported as per [XXXX] guideline (S1 Checklist)."

* When reporting 95% CIs please separate upper and lower bounds with commas instead of hyphens as the latter can be confused with reporting of negative values.

FIGURES AND TABLES

SUPPLEMENTARY MATERIAL

REFERENCES

OBSERVATIONAL STUDIES

* Abstract: We ask that Abstracts include the following information: the study design, population and setting, number of participants, years during which the study took place (enrollment and follow up), length of follow up, and main outcome measures. Could you kindly revise your Abstract to ensure all criteria are met, please?

* For all observational studies, in the manuscript text, please indicate: (1) the specific hypotheses you intended to test, (2) the analytical methods by which you planned to test them, (3) the analyses you actually performed, and (4) when reported analyses differ from those that were planned, transparent explanations for differences that affect the reliability of the study's results. If a reported analysis was performed based on an interesting but unanticipated pattern in the data, please be clear that the analysis was data driven. 

* Please state in the Methods section whether the study had a prospective protocol or analysis plan. If a prospective analysis plan (from your funding proposal, IRB or other ethics committee submission, study protocol, or other planning document written before analyzing the data) was used in designing the study, please include the relevant document(s) with your revised manuscript as a Supporting Information file to be published alongside your study and cite it in the Methods section. A legend for this file should be included at the end of your manuscript. If no such document exists, please make sure that the Methods section transparently describes when analyses were planned, and when/why any data-driven changes to analyses took place. Changes in the analysis, including those made in response to peer review comments, should be identified as such in the Methods section of the paper, with rationale.

---

## [Decision Letter · Decision Letter 2]

7 Jan 2025

Dear Dr. Fleetwood,

Thank you very much for re-submitting your manuscript "Depression and physical multimorbidity: a cohort study of physical health condition accrual in UK Biobank" (PMEDICINE-D-24-02561R2) for review by PLOS Medicine.

I have discussed the paper with my colleagues and the academic editor and it was also seen again by three of the original reviewers, whose comments are included below. I am pleased to say that provided the remaining editorial and production issues are dealt with, we plan to accept the paper for publication in the journal.

The remaining issues that need to be addressed are listed at the end of this email. Any accompanying reviewer attachments can be seen via the link below. Please take these into account before resubmitting your manuscript: [LINK]

A reminder that we ask every co-author listed on the manuscript to fill in a contributing author statement. If any of the co-authors have not filled in the statement, we will remind them to do so when the paper is revised. If all statements are not completed in a timely fashion this could hold up the re-review process. Should there be a problem getting one of your co-authors to fill in a statement we will be in contact.

We expect to receive your revised manuscript within 1 week. Please email me directly (hvanepps@plos.org) if you have any questions or concerns. Otherwise, we look forward to receiving the revised manuscript by Wed, Jan 15th, 

Kind regards,

Heather

Heather Van Epps, PhD

Executive Editor 

PLOS Medicine

hvanepps@plos.org

Requests from Editors:

1. Abstract, line 34-35 and line 36: Please refrain from expressing RRs as fold (or times)-increase; suggest modifying to “…participants with depression accrued physical morbidities at a faster rate than those without depression (RR 1.32…)…”

2. Abstract. Please include a sentence describing the main limitation(s) of the study at the end of the Methods and findings section.

3. For references to URLs (eg, ref 7, 21), please add the date accessed in brackets.

4. Please add a URL to the sponsor’s website to your funding statement.

5. Please add a URL for UK Biobank to your Data availability statement.

Comments from Reviewers:

Reviewer #1:

The authors have addressed my points

Michael Dewey

Reviewer #2: 

Authors have adequately addressed my comments and the paper has been much improved. I have no further comments.

Reviewer #4: 

All my comments have been thoughtfully and thoroughly addressed or clarified , and the manuscript is much improved overall. I do not have further recommendations to make and wish the authors success with this work.

[LINK]

---

## [Editor Report · Decision Letter 3]

13 Jan 2025

Dear Dr Fleetwood, 

On behalf of my colleagues and the Academic Editor, Vikram Patel, I am pleased to inform you that we have agreed to publish your manuscript "Depression and physical multimorbidity: a cohort study of physical health condition accrual in UK Biobank" (PMEDICINE-D-24-02561R3) in PLOS Medicine.

PRESS

Kind regards,

Heather

Heather Van Epps, PhD 

Executive Editor 

PLOS Medicine